# Step-size Optimization for Continual Learning

## Abstract

In continual learning, a learner has to keep learning from the data over its whole life time. A key issue is to decide what knowledge to keep and what knowledge to let go. In a neural network, this can be implemented by using a step-size vector to scale how much gradient samples change network weights. Common algorithms, like RMSProp and Adam, use heuristics, specifically normalization, to adapt this step-size vector. In this paper, we show that those heuristics ignore the effect of their adaptation on the overall objective function, for example by moving the step-size vector *away* from better step-size vectors. On the other hand, stochastic meta-gradient descent algorithms, like IDBD (Sutton, 1992), explicitly optimize the step-size vector with respect to the overall objective function. On simple problems, we show that IDBD is able to consistently improve step-size vectors, where RMSProp and Adam do not. We explain the differences between the two approaches and their respective limitations. We conclude by suggesting that combining both approaches could be a promising future direction to improve the performance of neural networks in continual learning.

## 1 The Role of Step-size in Continual Learning

Continual learning is a setting where learning needs to always adapt to the latest data to learn new things or track moving targets. Continual learning contrasts with other problem settings where the goal is to converge to some fixed point. A key problem in continual learning is to able to learn from data what needs to be maintained, for example to avoid catastrophic forgetting (French, 1999), and what needs to be updated to continue to track the objective function, for example because of limited capacity (Sutton et al., 2007).

A Stochastic Gradient Descent (SGD) method updates a set of weights $\boldsymbol{w}_t$ by incrementally accumulating the product of a step-size[1] scalar parameter $\eta$ and a sample gradient estimate $\widehat{\nabla J_t(\boldsymbol{w}_t)}$ given an objective function $J_t(\boldsymbol{w}_t)$:

$$\boldsymbol{w}_{t+1} \leftarrow \boldsymbol{w}_t - \eta\widehat{\nabla J_t(\boldsymbol{w}_t)}$$

We name this method Classic SGD in the rest of this paper. In Classic SGD, the step-size parameter $\eta$ is a key parameter to determine how much the weights of the learner are updated given the latest sample. For example, in a non-continual learning setting, this step-size parameter is often scheduled to converge to 0, forcing the changes to the parameters to be smaller over time until further adaptation becomes impossible. Using the same scalar step-size for all the weights is limited because it does not differentiate across dimensions.

Other conventional step-size adaptation methods include RMSProp (Hinton et al., 2012) and Adam (Kingma and Ba, 2015). RMSProp and Adam both normalize the gradients before updating the weights. Additionally, Adam also uses momentum to smooth the gradients. In practice, the weight update is composed of a component wise product of two parts: first, a step-size vector $\boldsymbol{\alpha}_{t+1}$ slowly changed and the gradient estimate $\widehat{\nabla J_t(\boldsymbol{w}_t)}$:

$$\boldsymbol{w}_{t+1} \leftarrow \boldsymbol{w}_t - \boldsymbol{\alpha}_{t+1} \cdot \widehat{\nabla J_t(\boldsymbol{w}_t)} \tag{1}$$

---

[1]We prefer to use *step-size* to *learning-rate* because we think it is more accurate. Indeed, a step-size parameter does not indicate the rate at which a system learns. A large step-size may or may not be correlated with a high rate of learning.

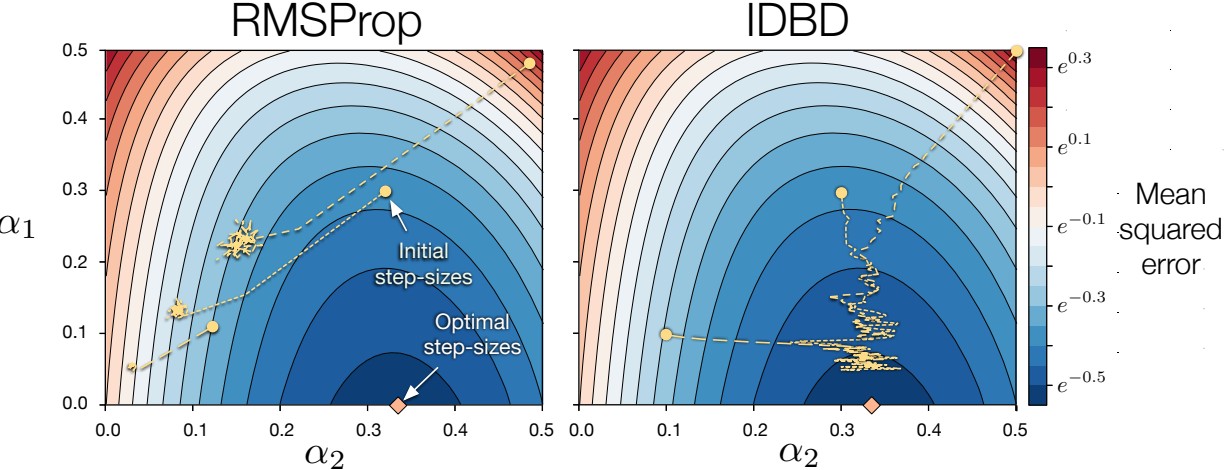

Figure 1: With conventional step-size normalization methods like RMSProp, the step-sizes do not go towards the optimal step-sizes.

Both RMSProp and Adam adapt the step-size vector $\boldsymbol{\alpha}_t$ using a heuristic, normalising with an estimate of the gradient magnitude, irrespective of the effect of such changes on the objective function. To illustrate this, we used a simplification of a problem introduced in Sutton (1992), that is, a non-stationary 2-dimensional linear regression problem. On the first dimension, the optimal weight $w_1$ is equals to 0 and constant. On the second dimension, the optimal weight $w_2$ is non-stationary. Every 20 steps, $w_2$ flips between -1 and 1 with a probability of 0.5. Features for both dimensions are independently sampled from a constant normal distribution. One would expect that a good method to optimize a step-size vector $(\alpha_1, \alpha_2)$ would learn a low step-size $\alpha_1$ for the first dimension to ignore noise, and learn a higher step-size $\alpha_2$ to track the flipping weight.

Figure 1 shows the loss landscape as the average squared error computed by running for 1,000,000 steps using linear regression as the objective function $J_t(\boldsymbol{w}_t)$, given the step-size vector at a point $(\alpha_1, \alpha_2)$, and using the update of Equation 1. Thus, Figure 1 is a representation of *the step-size space* with respect to the loss, and *not* the weight space as commonly depicted. In other words, each point represents how well regression is able to track the target given the fixed step-size vector at that point. Weights are initialized to 0. We can see that the pair of optimal step-sizes is $\alpha_1 = 0$ and $\alpha_2 \approx 0.33$, as indicated by the diamond at the bottom of the plots.

Figure 1-left shows the trajectory of the step-size vector $\boldsymbol{\alpha}_t$ in Equation 1 when updated by RMSProp using three different values for the step-size parameter. On this problem, we observe that RMSProp was not able to learn the best step-size vector to decrease the overall loss. Perhaps more surprisingly, in the two lower trajectories, the step-size vector ends up at a worse position from the optimal step-size vector compared to where it started from. Also note how RMSProp mostly moves both step-sizes $\alpha_1$ and $\alpha_2$ similarly, on a diagonal, and is not really able to distinguish between the different properties—a constant weight and a flipping weight—of the two dimensions. Although not shown in the figure, Adam behaves similarly to RMSProp on this problem.

Figure 1-right shows the trajectory of the step-size vector when updated by the Incremental-Delta-Bar-Delta (IDBD) algorithm (Sutton, 1992). IDBD is an online stochastic meta-gradient descent algorithm explicitly optimizing the step-size vector with respect to an objective function. More specifically, the IDBD algorithm learns a step-size vector by accumulating online estimates of the gradient of the objective function with respect to the step-sizes (IDBD is presented in detail later). Unlike RMSProp, IDBD is able to consistently update the step-size vector in the direction that is closer to the optimal point at the bottom of the landscape. Note that IDBD, RMSProp, and Adam all share the same compute and memory complexity.

We now explain this result in more details, highlighting the difference between step-size normalization algorithms (RMSProp, Adam) and step-size optimization algorithms (IDBD).

## 2   Setting

We consider an online learning setting where the learner observes a possibly non-stationary sequence $J_1, J_2, \ldots$ of loss functions. The learner starts with some initial weight vector $\boldsymbol{w}_1$ at time $t = 1$. At time $t = 1, 2, \ldots$, upon observing a new loss function $J_t(\cdot)$, the learner incurs a sample of the loss $J_t(\boldsymbol{w}_t)$ and then updates its weight vector $\boldsymbol{w}_t$ to $\boldsymbol{w}_{t+1}$ via some learning algorithm. The goal of the learner is to minimize the average lifetime loss.

Given a gradient sample $\nabla J_t(\boldsymbol{w}_t)$, a learner can use different algorithms to update its weight vector $\boldsymbol{w}_t$. Perhaps the simplest method is the Classic SGD algorithm introduced in Section 1 and presented below. Gradient samples $\nabla J_t(\boldsymbol{w}_t)$ are multiplied by a fixed scalar step-size parameter $\eta$ before being added to the current weight vector $\boldsymbol{w}_t$.

---

Classic Stochastic Gradient Descent (Classic SGD)

---

**Parameter:**
  $\eta$: step-size parameter for the weight update
Initialise weights $\boldsymbol{w}_1$
**for** $t = 1, 2, \ldots$ **do**
  $|\quad \boldsymbol{w}_{t+1} \leftarrow \boldsymbol{w}_t - \eta \nabla J_t(\boldsymbol{w}_t)$
**end**

---

Hinton et al. (2012) proposed the SGD algorithm named RMSProp, described below, which normalizes the gradient samples before they are accumulated in the weight vector. It does so with an additional vector $\boldsymbol{g}_t$ which maintains a component-wise moving average of the square of the gradient, $(\nabla J_t(\boldsymbol{w}_t))^2$. RMSProp introduces a new scalar step-size parameter $\gamma_g$ that sets how fast the average $\boldsymbol{g}_t$ tracks the square of the gradient. The average $\boldsymbol{g}_t$ is then used to normalise the update applied to the weight vector by dividing each component of the gradient by the square root $\sqrt{\boldsymbol{g}_t}$ of the average. A small constant $\epsilon$ is added to $\boldsymbol{g}_t$ for stability. Thus, the step-size vector $\boldsymbol{\alpha}_t$ for RMSProp becomes $\frac{\eta}{\sqrt{\boldsymbol{g}_t + \epsilon}}$.

---

RMSProp (Hinton et al., 2012)

---

**Parameters:**
  $\eta$: step-size parameter for the weight update
  $\gamma_g$: step-size parameter for normalization
  $\epsilon$: constant for numerical stability
Initialise weights $\boldsymbol{w}_1$
$\boldsymbol{g}_1 \leftarrow \mathbf{1}$
**for** $t = 1, 2, \ldots$ **do**
  $\Big|\quad \boldsymbol{g}_{t+1} \leftarrow (1 - \gamma_g)\boldsymbol{g}_t + \gamma_g\big(\nabla J_t(\boldsymbol{w}_t)\big)^2$
  $\Big|\quad \boldsymbol{w}_{t+1} \leftarrow \boldsymbol{w}_t - \frac{\eta}{\sqrt{\boldsymbol{g}_{t+1} + \epsilon}} \cdot \nabla J_t(\boldsymbol{w}_t)$
**end**

---

The Adam algorithm (Kingma and Ba, 2015), described below, adds two ideas to RMSProp. The first idea is *momentum*, where Adam replaces the gradient $\nabla J_t(\boldsymbol{w}_t)$ in the weight update with a tracking average of the gradient denoted $\boldsymbol{m}_t$. This tracking average is updated at every step given an additional step-size parameter denoted $\gamma_m$. Replacing the gradient with its average can be seen as a form of gradient smoothing. Setting $\gamma_m = 1$ disables momentum. The average of the gradient $\boldsymbol{m}_t$ and the squared gradient $\boldsymbol{g}_t$ are both initialized to zero. The second idea is to correct for the bias in the estimates of $\boldsymbol{g}_t$ and $\boldsymbol{m}_t$ because of that initialization to zero. Adam adjusts $\boldsymbol{g}_t$ and $\boldsymbol{m}_t$ to $\hat{\boldsymbol{m}}_t$ and $\hat{\boldsymbol{g}}_t$ respectively and uses these unbiased estimates in the weight update. The equivalent step-size vector $\boldsymbol{\alpha}_t$ for Adam is $\frac{\eta}{\sqrt{\hat{\boldsymbol{g}}_t + \epsilon}}$.

---

Adam (Kingma and Ba, 2015)

---

**Parameters:**

  $\eta$: step-size parameter for the weight update
  $\gamma_m$: step-size parameter for gradient smoothing
  $\gamma_g$: step-size parameter for normalization
  $\epsilon$: constant for numerical stability

Initialise weights $\boldsymbol{w}_1$

$\boldsymbol{m}_1 \leftarrow \boldsymbol{0}$

$\boldsymbol{g}_1 \leftarrow \boldsymbol{0}$

**for** $t = 1, 2, \ldots$ **do**

  $\boldsymbol{m}_{t+1} \leftarrow (1 - \gamma_m)\boldsymbol{m}_t + \gamma_m \nabla J_t(\boldsymbol{w}_t)$

  $\boldsymbol{g}_{t+1} \leftarrow (1 - \gamma_g)\boldsymbol{g}_t + \gamma_g \big(\nabla J_t(\boldsymbol{w}_t)\big)^2$

  $\hat{\boldsymbol{m}}_{t+1} \leftarrow \frac{\boldsymbol{m}_{t+1}}{1 - (1 - \gamma_m)^t}$

  $\hat{\boldsymbol{g}}_{t+1} \leftarrow \frac{\boldsymbol{g}_{t+1}}{1 - (1 - \gamma_g)^t}$

  $\boldsymbol{w}_{t+1} \leftarrow \boldsymbol{w}_t - \frac{\eta}{\sqrt{\hat{\boldsymbol{g}}_{t+1} + \epsilon}} \cdot \hat{\boldsymbol{m}}_{t+1}$

**end**

---

## 3   Limitations of Step-size Normalization

This section highlights the limitations of step-size normalization on two simple learning problems, namely *weight-flipping* and *rate-tracking*. For both problems, we examine a simple setting with a linear least mean squared regression loss function. For $t = 1, 2, \ldots$, the losses $J_1, J_2, \ldots$ are of the form $J_t(\boldsymbol{w}_t) = (\boldsymbol{x}_t^\intercal \boldsymbol{w}_t - y_t^*)^2$, where $\boldsymbol{x}_t \in \mathbb{R}^d$ is the feature vector and $y_t^* \in \mathbb{R}$ is the label at time $t$. In such setting, $\nabla J_t(\boldsymbol{w}_t) = \delta_t \boldsymbol{x}_t$ where $\delta_t = \boldsymbol{x}_t^\intercal \boldsymbol{w}_t - y_t^*$.

**The weight-flipping problem.**

The problem was first introduced in Sutton (1992) and is a larger version of the problem introduced in Section 1. It defines a vector $\boldsymbol{x}_t$ of 20 inputs where each input is sampled independently according to a normal distribution with zero mean and unit variance. The first 15 components of the target weight vector $\boldsymbol{w}_t^*$ are all zeros. The last 5 components are either +1 or -1. To make it a continual learning problem, one of the non-zero weights is selected every 20 samples and flipped from +1 to -1 or vice-versa. Finally, the target prediction $y^*$ is defined as the linear combination of the weight vector and the input $y^* = \boldsymbol{w}_t^{*\intercal} \boldsymbol{x}_t$.

In this problem, a performance close to the optimal performance can be obtained by learning a different but constant step-size per component; that is, a learning algorithm able to learn a low step-size for the first 15 constant weights and a high step-size for the 5 flipping weights will perform better than an algorithm using the same step-size for all weights.

The asymptotic performance of Classic SGD, RMSProp, and Adam on the weight-flipping problem as a function of the step-size parameter is shown in Figure 2. We treated RMSProp as a special case of Adam where $\gamma_m = 1$ because we considered that the problem is run for long enough to clear any bias due to initialisation. We did a sweep over $\gamma_g$ and $\eta$. The performance for the best values of $\gamma_g$ and $\eta$ are reported in the figure. As an additional baseline, we ran Oracle SGD where the weights and step-sizes for the first 15 constant components are set to an optimal value of 0 and ran a sweep over the step-size for the remaining non-constant last 5 components. Consequently, Oracle SGD shows the best performance possible with a constant vector step-sizes in this problem setting. We report the error averaged across all steps after running for 200,000 steps.

On this problem, Adam/RMSProp performed slightly worse than Classic SGD and much worse than Oracle SGD. Their poor performance can be explained by looking at the algorithms. Adam/RMSProp maintains an average of $\delta_t \boldsymbol{x}_t^2$; however, because all components of $\boldsymbol{x}_t$ are from the same normal distribution, they have the same variance. Moreover, because the error $\delta_t$ is global, all components end up with the same normalised step-size estimate $\frac{\eta}{\sqrt{\boldsymbol{g}_t + \epsilon}}$. Consequently, on such tracking problem, Adam/RMSProp is not able to learn

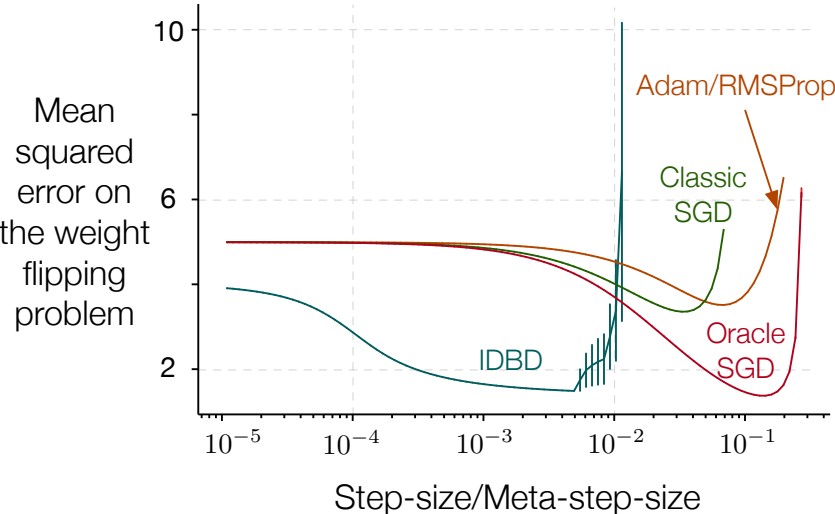

Figure 2: On the weight-flipping problem, IDBD performs as well as Oracle SGD and better than conventional methods.

different step-sizes for each component. We conclude that normalization, as done in Adam/RMSProp, is not enough to differentiate between weights that should be fixed and weights that should change.

**The 1D noisy rate-tracking problem.**

In the previous problem, learning constant step-size parameters lead to an optimal solution. Generally, in a continual learning setting, the optimal step-sizes may not be constant and may need to be adapted over time. The noisy-tracking problem illustrates such a case. This problem is still a linear regression problem of a single dimension where the feature $x_t$ equals 1, at all times $t$.

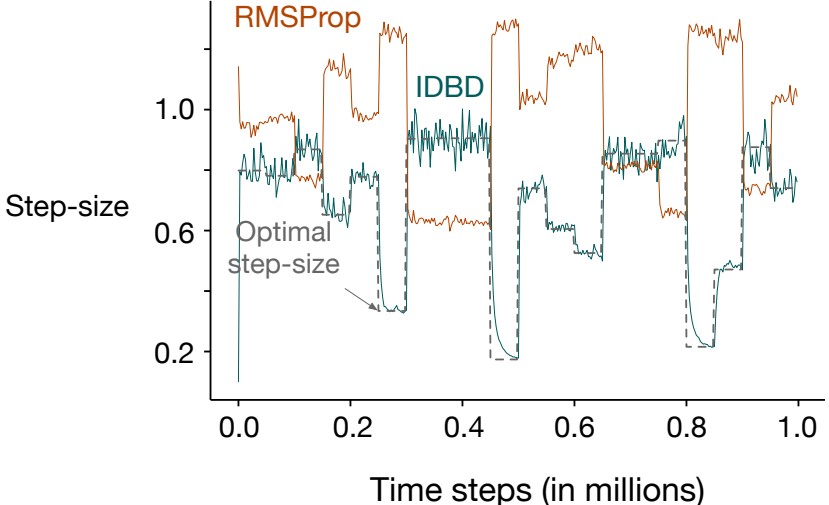

Figure 3: On the noisy-tracking problem, step-size optimization (IDBD) can accurately track the optimal step-size on a non-stationary single dimension problem. Step-size normalization, as done by RMSProp, on the other hand, achieves exactly the opposite—it increases the step-size when it should be decreased and vice-versa.

The learner aims to predict the target $y$ that changes at every step as:

$$y_t = z_t + \mathcal{N}(0, \sigma_t^2)$$
$$z_{t+1} = z_t + \mathcal{N}(0, 1)$$

where $\sigma_t$ is sampled every 50,000 steps from a uniform distribution from 0 to 3. Sutton (1981) showed that the optimal step-size $\alpha_t^*$ for this problem can be computed as:

$$\alpha_t^* = \frac{-\sigma_t^2 + \sqrt{\sigma_t^4 + 4\sigma_t^2}}{2}. \tag{2}$$

Figure 3 shows that Classic SGD and RMSProp are not able to track the optimal step-size $\alpha_t^*$ on the 1D noisy rate-tracking problem. Indeed, because $x_t$ is always equals to 1, RMSProp will decrease the step-size every time the magnitude of the error increases. However, the opposite is needed for this problem. The increase in the magnitude of the error is due to a faster rate of drift in the target, which requires a larger step-size. Figure 3 shows that RMSProp does the opposite of what needs to be done: increases the step-size when it should be decreased and vice-versa.

The experimental protocol is similar to the weight-flipping problem. We ran the experiment for 1 million steps. The trajectory reported in Figure 3 is for Adam with $\gamma_m = 1$, where momentum is disabled, which is why we labeled the algorithm RMSProp. As before, we consider the correction of the bias at the start of the trajectory to be negligible. Results are reported for the best hyper-parameter configuration.

## 4   Step-size optimization with IDBD

The Increment-Delta-Bar-Delta (IDBD) algorithm (Sutton, 1992), presented next, is a step-size optimization method. Like RMSProp and Adam, IDBD is a two-level learning process: a *base* level and a *meta* level. The base level learns the weight vector $\boldsymbol{w}_t$ used in the loss $J_t(\boldsymbol{w}_t)$. The meta level learns *meta-parameters*, that is the step-size vector $\boldsymbol{\alpha}_t$ in Equation 1, used in the base level to update the weights $\boldsymbol{w}_t$. The key difference between a step-size optimization method (e.g. IDBD) and a step-size normalization method (e.g. RMSProp) is that a step-size optimization method has an update rule for the meta level optimizing for the *same* loss $J_t(\cdot)$ than the base level, as opposed to a step-size normalization method that uses a heuristic, normalization, to update the step-size vector $\boldsymbol{\alpha}_t$ at the meta level.

---

**Increment-Delta-Bar-Delta (IDBD) (Sutton, 1992)**

**Parameters:**
       $\alpha_0$: initial step-size
       $\theta$: meta step-size for the step-size update
Initialise weights $\boldsymbol{w}_1$
Initialise step-sizes in log space $\boldsymbol{\beta}_1 = \boldsymbol{log}(\alpha_0)$
Initialise average vector $\boldsymbol{h}_1 = \boldsymbol{0}$
**for** $t = 1, 2, \dots$ **do**
    $y_t \leftarrow \boldsymbol{w}_t^\mathsf{T}\boldsymbol{x}_t$
    $\delta_t \leftarrow y_t^* - y_t$
    $\boldsymbol{\beta}_{t+1} \leftarrow \boldsymbol{\beta}_t + \theta \delta_t \boldsymbol{x}_t \cdot \boldsymbol{h}_t$
    $\boldsymbol{\alpha}_{t+1} \leftarrow e^{\boldsymbol{\beta}_{t+1}}$
    $\boldsymbol{w}_{t+1} \leftarrow \boldsymbol{w}_t + \boldsymbol{\alpha}_{t+1} \cdot \delta_t \boldsymbol{x}_t$
    $\boldsymbol{h}_{t+1} \leftarrow \left(1 - \boldsymbol{\alpha}_{t+1} \cdot \boldsymbol{x}_t^2\right)^+ \cdot \boldsymbol{h}_t + \boldsymbol{\alpha}_{t+1} \cdot \delta_t \boldsymbol{x}_t$
**end**

---

To optimize at the meta level for the same loss than the base level, the general idea of meta-gradient proposes to derive the update rule of the meta level by taking the gradient of the loss with respect to the meta-parameters used in the update of the base level. In the case of IDBD, the step-size vector is defined as $\boldsymbol{\alpha}_t = e^{\boldsymbol{\beta}_t}$, that is the exponential of a vector $\boldsymbol{\beta}_t$. Updating the step-sizes in log-space guarantees that $\boldsymbol{\alpha}_t$

is always positive, and enables fast geometric updates. Other representation could have been chosen, like a sigmoid for example. Consequently, the IDBD algorithm is derived by taking the gradient of the linear least mean squared regression loss $J_t(\boldsymbol{w}_t) = (\boldsymbol{x}_t^\intercal \boldsymbol{w}_t - y_t^*)^2$ with respect to the step-size vector $\boldsymbol{\beta}_t$ in log-space. Additionally, IDBD uses approximations in its derivation to keep the same complexity than Classic-SGD. See Sutton (1992) for a complete description of that derivation.

We now give a mechanistic description of the IDBD algorithm to explain how it works. As with previous methods, IDBD first computes the error $\delta_t$. IDBD also keeps an average $\boldsymbol{h}_t$ of the gradient $\delta_t \boldsymbol{x}_t$ for each component of the input $\boldsymbol{x}_t$. For a component $i$ at time $t$, the absolute value of the average $\boldsymbol{h}_{(t,i)}$ will be high for a weight $\boldsymbol{w}_{(t,i)}$ if the recent gradients $\delta_t \boldsymbol{x}_{(t,i)}$ always have the same sign (moving in the same direction) and low if recent gradient oscillates around 0. Then, the step-size in log space $\boldsymbol{\beta}_{(t,i)}$ is increased if the last gradient $\delta_t \boldsymbol{x}_{(t,i)}$ correlates with the average $\boldsymbol{h}_{(t,i)}$, and decreased otherwise. Consequently, the step-size $\boldsymbol{\alpha}_{(t,i)}$ increases if the weight $\boldsymbol{w}_{(t,i)}$ moves in a consistent direction over time (meaning the step-size was too small) and decreases when the direction oscillates (meaning the step-size was too high). The notation $(\cdot)^+$ guarantees that $\left(1 - \boldsymbol{\alpha}_{t+1} \cdot \boldsymbol{x}_t^2\right)^+$ is between 0 and 1 to discount the previous value $\boldsymbol{h}_t$.

The performance of IDBD on the weight-flipping and rate-tracking problems is shown on Figure 2 and Figure 3 respectively. On both cases, IDBD outperformed Classic SGD, RMSProp, and Adam. On the weight-flipping problem, IDBD learns to decrease the step-sizes for the inputs with constant weights and to keep the step-size high for inputs with flipping weights. On the 1D noisy rate-tracking problem, IDBD is able to track the optimal step-size.

## 5 Limitations of IDBD

The next step would be to use IDBD to learn the step-sizes in deep neural networks. A few open questions remain to be solved for such an endeavor. The first is how to generalize IDBD such that it be used to optimize all hyper-parameters (See Section A for a discussion). The second is that the main parameter of IDBD, the meta-step-size hyper-parameter, is sensitive to the magnitude of the gradients as shown in Figure 4. In addition to the weights flipping between -1 and +1, labelled "Medium" target weights, we also run a "Small" target weights and a "Large" target weights variants, respectively flipping from -.1 and +.1 and -10 and +10. Other parameters are the same. These two variants shifted the best meta-step-size for IDBD by 5 orders of magnitudes. Classic SGD on the other hand did not shift. Such sensitivity of the meta-step-size parameter to the variance of the inputs, the gradients, or the updates in general, makes it difficult to use IDBD in many common settings.

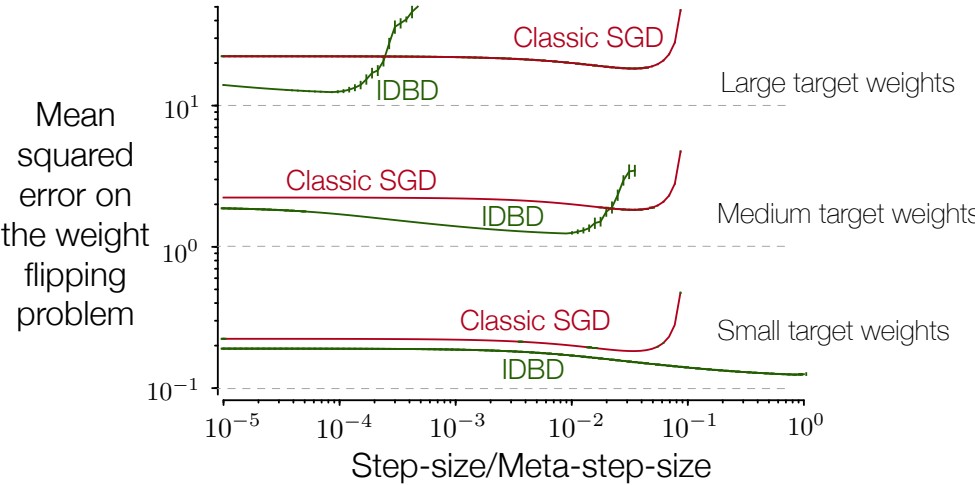

Figure 4: Shift of the best meta step-size parameter of the IDBD algorithm.

## 6    Existing Extensions of IDBD

IDBD was originally designed for linear regression but the idea of stochastic meta-gradient descent introduced in the paper is general. Indeed, a general formulation of meta-gradient in online optimization problems has been derived in Xu et al. (2018) and is detailed in Appendix A. Meta-gradient in general has been used in a wide variety of settings, see Andrychowicz et al. (2016) or Xu et al. (2020) for examples.

There are different extensions of the IDBD algorithm, more specifically to different learning settings and problems. For example, Koop (2008) extends IDBD to classification problems over linear regression problems. Kearney et al. (2019), Thill (2015), and Young et al. (2018) extended the IDBD algorithm for temporal-difference (TD) learning in a reinforcement learning context. Schraudolph (1999) extends the IDBD algorithm to multi-layer neural network with the SMD algorithm. SMD has been later applied to areas like independent component analysis (Schraudolph and Giannakopoulos, 2000) and complex human motion tracking (Kehl and Van Gool, 2006).

To address the sensitivity of IDBD and SMD to the meta-step-size parameter, Mahmood et al. (2012) proposed the Autostep algorithm as a tuning-free extension. Sutton (2022) summarizes the history and recent success of meta-learning algorithms in step-size adaptation methods.

## 7    Other Step-size Adaptation Algorithms

There are numerous optimization techniques developed for improving convergence or tracking of optimal parameters in stationary or online optimization problems (Amari et al., 2000; Roux and Fitzgibbon, 2010; Schaul et al., 2013; Desjardins et al., 2015; Bernacchia et al., 2018). These techniques include (but are not limited to) second order optimization methods, variance reduction methods, and step-size normalization techniques. General meta-gradient step-size updates, like IDBD, are conceptually different from the aforementioned methods. For example, second order optimization methods such as Newton and quasi-Newton methods leverage information about the curvature of the loss landscape in weights space to obtain an improved direction (Kochenderfer and Wheeler, 2019). In contrast, meta-gradient algorithms like IDBD perform updates in hyper parameters space (such as step-sizes) instead of the weight space (as demonstrated in Figure 1). Note also that in the example problems of Section 3, second order methods have almost no advantage over first order methods because the average curvature in these problems is independent of the weights and is almost constant over different dimensions.

Other approaches include the work of van Erven and Koolen (2016), which introduces an online algorithm that runs multiple sub-algorithms, each with a different step-size, and learns to use the best step-size for online convex optimization problems. Wu et al. (2020) proposes to adapt the step-size according to accumulating gradients for L-Lipschitz continuous objectives. Koolen et al. (2014) proposes to use a grid of step-size and runs in linear time for prediction problems with expert knowledge. Jacobsen et al. (2019) introduces AdaGain. AdaGain was designed to work alongside other step-size adaptation methods. To that end, Adagain optimizes a generic proxy objective function independent of the base objective function: the vector of step-size is optimized so that the norm of the gradient of the base objective function becomes small. They show that AdaGain with RMSProp, for example, can outperform other methods in a continual learning setting.

Finally, variance reduction techniques such as SVRG (Johnson and Zhang, 2013) and SAGA (Defazio et al., 2014) try to smooth out the noise and obtain lower-variance approximations of gradient of expected loss in stationary problems. This limits the application of these methods to online learning problems where such expected loss function does not exist.

## 8    A Promising Research Direction: Normalized Step-size Optimization

Adam and RMSProp are step-size normalization methods widely used in deep learning. They use momentum and normalization techniques to adapt the step-size vectors, often providing more stable updates compared to Classic SGD. These methods have been successful and are now ubiquitous. This paper has shown that they may not be enough, in the context of continual learning. On the other hand, meta-gradient algorithms

like IDBD optimize the step-sizes with respect to the loss, but have other limitations such as stability and sensitivity to its step-size parameter. Optimizing step-sizes in deep neural-networks in a practical way for continual learning is still an open research question. Consequently, we see that combining normalization and optimization seem a promising research direction towards better step-size adaptation methods in deep networks. The Autostep algorithm (Mahmood et al., 2012) can be seen as a possible attempt towards such direction.

Finally, it is possible that a good step-size adaptation method will improve learning in the continual learning setting but also in other setting. For example, a good step-size adaptation method may remove the need for a manually tuned step-size schedule, having to sweep over constant step-size parameters, or better learning in long training of large models.

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

---

Generalisation of the IDBD Algorithm for Online Learning Problems

---

**Parameters:**

$\theta$: meta step-size for the step-size update

Initialise parameters $\boldsymbol{w}_1$
Initialise step-sizes in log space $\boldsymbol{\beta}_1$
Initialise average vector $\boldsymbol{h}_t = \boldsymbol{0}$
**for** $t = 1, 2, \ldots$ **do**

$\quad \boldsymbol{\beta}_{t+1} \leftarrow \boldsymbol{\beta}_t + \theta \boldsymbol{h}_t \nabla J_t(\boldsymbol{w}_t)$
$\quad \boldsymbol{\alpha}_{t+1} \leftarrow \exp(\boldsymbol{\beta}_{t+1})$
$\quad \boldsymbol{w}_{t+1} \leftarrow \boldsymbol{w}_t - \boldsymbol{\alpha}_{t+1} \nabla J_t(\boldsymbol{w}_t)$
$\quad \boldsymbol{h}_{t+1} \leftarrow (1 - \boldsymbol{\alpha}_{t+1}\boldsymbol{g}_t)^+ \boldsymbol{h}_t + \boldsymbol{\alpha}_{t+1}\nabla J_t(\boldsymbol{w}_t), \qquad$ where $\boldsymbol{g}_t$ is defined in (A)

**end**

---

## A   Generalisation of IDBD

An extension of IDBD for meta-gradient update of hyper parameters has been derived in Xu et al. (2018). That general method to step-size adaptation can be seen as an extension of IDBD for step-size adaptation in general online learning problems.

For an objective function $J_t$, the meta-gradient update of step-sizes is as follows (Xu et al., 2018):

$$\boldsymbol{\beta}_{t+1} \leftarrow \boldsymbol{\beta}_t - \theta \boldsymbol{H}_t^{\mathsf{T}} \nabla J_t(\boldsymbol{w}_t)$$
$$\boldsymbol{A}_{t+1} \leftarrow \mathrm{diag}\big(\exp(\boldsymbol{\beta}_{t+1})\big),$$
$$\boldsymbol{w}_{t+1} \leftarrow \boldsymbol{w}_t - \boldsymbol{A}_{t+1}\nabla J_t(\boldsymbol{w}_t),$$
$$\boldsymbol{H}_{t+1} \leftarrow \big(I - \boldsymbol{A}_{t+1}\nabla^2 J_t(\boldsymbol{w}_t)\big) \boldsymbol{H}_t - \mathrm{diag}\big(\boldsymbol{A}_{t+1} \nabla J_t(\boldsymbol{w}_t)\big),$$

where $\mathrm{diag}(\boldsymbol{x}_t)$ stands for a diagonal matrix with diagonal entries equal to the entries of vector $\boldsymbol{x}_t$, and $\nabla^2 J_t(\boldsymbol{w}_t)$ is the Hessian matrix of $J_t$ at $\boldsymbol{w}_t$. As opposed to the average vector $\boldsymbol{h}_t$ in the IDBD algorithm, here the average $\boldsymbol{H}_t$ is a square matrix, and its update requires $\Theta(d^2)$ computations. To obtain an $O(n)$-complexity algorithm, we can consider a diagonal approximation of $\nabla^2 J_t(\boldsymbol{w}_t)$. In particular, at time $t$, we let $\boldsymbol{g}_t$ be a $d$-dimensional vector with entries

$$g_{(t,i)} = \frac{d^2}{dw_i^2} J_t(\boldsymbol{w})\big|_{\boldsymbol{w}=\boldsymbol{w}_t}, \qquad \text{for } i = 1, \ldots, d.$$

The resulting generalized IDBD algorithm with a linear-complexity is in the pseudo-code. It is then easy to check that in the linear regression problem, this algorithm simplifies to the IDBD described in Section 4 of the paper.

