# OpenReview forum: "Step-size Optimization for Continual Learning"
_TMLR — Rejected by TMLR_

### Review · Reviewer_AgGe · 2023-09-12

**Summary Of Contributions:**

This paper considers the continual learning problem from the optimization perspective. In particular, it discusses modern optimizers such as RMSProp and Adam and their inefficiency in certain continual learning settings with changing data distribution. As a possible improvement, the paper suggests the method IDBD, which was proposed by Sutton in 1992. The authors compare the optimizers on several toy problems, including the weight-flipping problem and the noisy rate-tracking problem. They show that Adam and RMSProp are incapable of finding the correct step sizes in these settings and as such they are often outperformed even by simple SGD, while IDBD works well.

**Audience:**

No

**Claims And Evidence:**

Yes

**Requested Changes:**

There are numerous interesting directions to extend this work, however, I don't think these are viable in the TMLR review process timeframe. The most obvious example would be to extend the study to consider some more realistic problems, basically, anything studied in the current CL literature, e.g. split-CIFAR, CORe50, NEVIS'22, Continual World, sequential Atari. Another way to extend this work would be a thorough survey of optimizers in CL settings or provide a comprehensive theoretical framework for studying adaptive step size in CL problems.

**Strengths And Weaknesses:**

Unfortunately, although this paper touches on an important problem, as of now it seems like a preliminary work mostly recapitulating results already present in the literature. I don't think there are enough insights as of yet for TMLR standards, but I'd be interested to see an extended version of this study that digs deeper into the stated problem.

Strengths:
- The paper considers an important problem of optimization in the continual learning setting and approaches it in a principled way.
- The paper is clearly written and the thought process is easy to follow.
- The related work is discussed sufficiently, in particular, the authors discuss several older works, not limiting themselves to only the most recent papers. This is certainly a good practice.


Weaknesses:
- The paper offers little to no novelty. For TMLR, the novelty is not the primary criterion for the paper to be accepted, but I find almost no new observations here, the paper mostly recapitulates results from previous works. As such, I'm not sure if this work would be interesting for the TMLR community.
    - The paper notes that popular optimizers such as RMSProp and Adam underperform on certain toy continual learning problems. I would say that this is the only truly novel observation but it's still not very surprising given that the CL community observed problems with such optimizers in the past [1, 2].
    - The paper shows that meta-gradient approaches such as IDBD proposed by Sutton in 1992 work well in toy settings, some of which were proposed in the same paper. I would be interested to see these experiments extended to more complex benchmarks and models, such as neural networks, but the paper does not consider any modern continual learning problem. As the authors note, there are certain limitations of IDBD that make it difficult to scale it up, but then one could try to use some of the methods from the meta-gradient family.
    - Finally, although Adam is still the most popular optimizer the study would gain from considering different types of optimizers that have been proposed over the years.
    - Additionally, as the authors observe the IDBD method isn't actually forgotten and methods that are similar in spirit are still present today e.g. in the meta-gradient literature. So I also don't think there is a contribution in terms of finding a lost paper that would be interesting for the community today.

Minor Comments:
- "We prefer to use step-size to learning-rate because we think it is more accurate" - in general I agree that it's more accurate, but I would say that learning rate is by now a standardized term in machine learning so I would suggest leaving it at that.
- The paper doesn't consider the role of initialization in the training process, e.g. I don't think that the initialization scheme for Figure 1 is specified anywhere. At some point, the authors write: "We treated RMSProp as a special case of Adam where γm = 1 because we considered that the problem is run for long enough to clear any bias due to initialisation". I would appreciate a more thorough discussion of this, as I don't think this statement would hold true for deeper models. With linear ones, it's possible that the impact is indeed negligible but I think this should be shown theoretically or empirically.
- "Note that IDBD, RMSProp, and Adam all share the same compute and memory complexity" - isn't RMSProp cheaper as it does not track momentum? I agree that the order is the same, but there are some differences that might make an impact in practice.
- There are some grammar errors:
    - "A key problem in continual learning is to able"
    - "the optimal weight w1 is equals to 0"
    - the paper mixes British and American English, e.g. "initialisation" and "initialization".


[1] Hsu, Yen-Chang, et al. "Re-evaluating continual learning scenarios: A categorization and case for strong baselines." arXiv preprint arXiv:1810.12488 (2018). \
[2] Lesort, Timothee, et al. "Challenging Common Assumptions about Catastrophic Forgetting." (2022).

---

> ### Author Response · Authors · 2023-09-18
> **Response to reviewer AgGe**
>
> We thank you very much reviewer AgGe for his detailed comments. You are pointing out that the main weakness of our paper is that there are not enough insights for TMLR standards, specifically that there are “no new observations” in the paper. It is true that the poor performance of RMSProp/Adam in the CL framework has been observed in the past (thank you for the pointers to the papers). The main contribution of our paper is to explain **WHY** it is so. The two papers cited in your review did observe the poor performance of RMSProp/Adam but give no explanation of why. Identifying the root cause of a problem seems to us a necessary first step to solve that problem. To that end, this is why our paper has focused on simple settings to isolate and *explain* the core problem as much as possible.
>
> As an extension, you suggest considering more realistic problems, like benchmarks previously introduced in the literature, writing a survey of optimisers in CL settings, or providing a comprehensive CL theoretical framework. It is unclear to us how such extensions would help explaining the root cause of why RMSProp/Adam fails in the CL setting and what to do about it, like our paper does. Please also note that solving this core problem could have an impact beyond the CL setting.
>
> Finally, it is true that the root cause is simple once identified. Given the popularity of Adam and its variants, we think it is an important true fact worth its own paper. We also think this observation is new because we could not find any paper explaining why RMSProp/Adam have limitations in the CL setting, for example mentioning the adversary effects of the gradient normalisation. Please let us know if you think we have missed papers on that topic.
>
> Response to minor comments:
> * you wrote “_one could try to use some of the methods from the meta-gradient family_”: recent works about meta-gradient are not applicable in the setting of the paper. Indeed, recent works have mostly focused on optimising few hyper-parameters often in an outer loop of learning, as opposed to per-weight step-size parameters optimised at every update. We are not aware of any recent online IDBD-like meta-gradient algorithms. In that sense, we think IDBD and SMD have indeed been forgotten. Additionally, extending IDBD to the non-linear case and making it robust is still an open question. We think that including such work in this paper would be distracting from the main goal of the paper. We will clarify this in the paper.
> * you wrote “_the study would gain from considering different types of optimisers”: most modern optimisers (Novograd, Yogi, AdamaxW, SM3, …) are variations of Adam and suffer the same limitations identified in our paper. We will mention this in the paper. Please let us know if you think we have missed an optimiser that would not be in the Adam’s family (puns intended :D).

---

> > ### Comment · Reviewer_zMmu · 2023-10-03
> >
> > Hello, jumping in this thread preemptively
> >
> > > The main contribution of our paper is to explain WHY it is so
> >
> > I don't think this is a fair claim. Your paper does not provide a theory of divergence in CL, i.e. not a *why*, it shows the reader two examples in which you provide an observational account of the divergence phenomena. It remains specific to the two examples you've chosen.

---

> > > ### Author Response · Authors · 2023-10-04
> > > **Answer to zMmu**
> > >
> > > Yes, we do not provide a theory of divergence in CL because it is unlikely to exist. Instead, we provide meaningful counter-examples. Indeed, the simplicity of our counter-examples make them general and relevant to a wide variety of settings. Specifically the problem in Figure 3 which can be seen as a model of what learning is about in its simplest form --- tracking a moving scalar target.
> > >
> > > We also spend quite some time in the paper, in Section 3 specifically, to explain that RMSProp/Adam simply do not track the right quantities to do well in the CL setting. Still in section 3, we also explain why these algorithms cannot make the difference between the error increasing because of the noise or because the target is moving, which is a core feature in a CL setting.
> > >
> > > To summarise, it is unclear to us what better explanation of "why" we could provide to explain the current performance of Adam/RMSProp in the CL setting.

---

### Review · Reviewer_2Umh · 2023-09-18

**Summary Of Contributions:**

This paper proposes an experimental comparison between gradient descent algorithms such as Adam and RMSProp, and a gradient descent algorithm with trained learning rate, IDBD. The study focuses on linear models with a goal changing over time. According to the experimental results, IDBD is able to learn the optimal step size (for each parameter), while Adam and RMSProp fail.

**Audience:**

No

**Claims And Evidence:**

Yes

**Requested Changes:**

It would be interesting to propose a wider experimental setup, involving deep neural networks, where Adam is known to perform well, and extensions of IDBD (cited in Section 6). Specifically, many practitioners would be interesting in a description of the failure modes of Adam (in RL), and alternative methods to avoid them.

I do not think that such a complex extension of the work could be done in a quick revision of the present paper.

**Strengths And Weaknesses:**

# Strengths

The paper is well-written, sheds a new light on the situations where Adam and RMSProp are supposed to work. Also, it is a reminder on an existing meta-learning method (IDBD) that has some advantages in certain circumstances (varying goal).

# Weaknesses

The main weakness is the significance of the work: the methods and the experimental setups exist already. Moreover, this experimental setup is restricted to the linear regression.

---

### Review · Reviewer_zMmu · 2023-10-02

**Summary Of Contributions:**

This paper analyses SGD, RMSProp and IDBD in some non-stationary settings, showing that the commonly used Adam/RMSProp optimizers are not suited to the task. A short experiment also shows a basic limitation of IDBD to input scale. This is accompanied by an overview of prior work in metagradient and non-stationary optimization.

**Audience:**

No

**Broader Impact Concerns:**

No concerns

**Claims And Evidence:**

Yes

**Requested Changes:**

I do feel that the basis of this paper, the comparison with RMSProp and Adam [a] is awkward, in the sense that they are not designed in the first place to be robust to non-stationarity. I agree that that's what people use in practice and so we should depart from that, but that's a very different argument than "look this doesn't do what it's not designed to!". I'm sure 95% of people using Adam in CL know that it's not made for that. There are also modern extensions of IDBD, which already perform similar work.

In that sense, the paper really feels like it's underdelivering. If this was the build-up to the introduction of a novel optimization algorithm it would be great, but it's not.
The bar is relatively low for TMLR, the main one is correctness, and this paper is correct, as far as I'm able to judge. Let's look at the other criteria:
- new algorithms with sound empirical validation, optionally with justification of theoretical, psychological, or biological nature;
- experimental and/or theoretical studies yielding new insight into the design and behavior of learning in intelligent systems;
- accounts of applications of existing techniques that shed light on the strengths and weaknesses of the methods;
- formalization of new learning tasks (e.g., in the context of new applications) and of methods for assessing performance on those tasks;
- development of new analytical frameworks that advance theoretical studies of practical learning methods;
- computational models of natural learning systems at the behavioral or neural level;
- reproducibility studies of previously published results or claims;
- new approaches for analysis, visualization, and understanding of artificial or biological learning systems.

Maybe this paper satisfies the second category, but IMHO fails to meet the bar for novel insights. I don't feel like I've learned anything of substance in this paper. In fact, the section relating to existing stochastic meta-gradient descent methods quite nicely shows that this is an old and well understood problem. I think this is an important problem worthy of study, but I think the authors could do more here to build a much better paper.

[a] RMSProp and Adam work because they can escape saddle points [1] in high-dimensional optimization. I wouldn't really expect them to work in any other setting, in fact there's plenty of work suggesting they're quite "broken" in RL [2,3].

Not asking the authors to cite those papers, just there to support my arguments:
[1] Identifying and attacking the saddle point problem in high-dimensional non-convex optimization, Yann Dauphin, Razvan Pascanu, Caglar Gulcehre, Kyunghyun Cho, Surya Ganguli, Yoshua Bengio, 2014
[2] The Primacy Bias in Deep Reinforcement Learning, Evgenii Nikishin, Max Schwarzer, Pierluca D’Oro, Pierre-Luc Bacon, Aaron Courville, 2022
[3] Correcting Momentum in Temporal Difference Learning, Emmanuel Bengio, Joelle Pineau, Doina Precup, 2021


Other remark, the description of IDBD is a bit too simplistic: "the IDBD algorithm is derived by taking the gradient of the linear least mean squared regression loss [equation] with respect to the step-size vector βt in log-space." This would confuse anyone not familiar with IDBD, for one $\beta$ does not appear anywhere in that equation. In fact the original derivation by Sutton requires us to assume the existence of some metagradient parameters $\mathbf{h}$ for the substitution to make sense in the derivative taken with respect to the _update_ (not the loss $J$).

**Strengths And Weaknesses:**

The paper is an interesting tutorial for readers unfamiliar with the problem. It distills into the simplest examples the reason for meta-gradient and non-stationary methods to exist.

The main weakness of the paper is that its contribution is quite limited. It exposes an already known problem, sure in a didactic way, but falls short of a review paper. It's not clear to me how this paper is advancing the discussion.

---

### Decision · Action_Editor_qaar · 2023-11-06

**Recommendation:** Reject

**Comment:**

All three reviewers agree, that in its current status, the paper is not ready to be published at TMLR, as summarized above.

I encourage the authors to take the reviewers' comments, suggestions, and references into serious consideration, to incorporate them, and then to resubmit a paper that will be a lot stronger.

**Audience:**

The paper lacks a clear motivation why the failure of optimizers designed for high-dimensional nonlinear objectives is relevant, and what deeper insight is provided beyond what has already been noted in prior work. (See references within the reviews.)
As such we cannot expect it to be of interest and benefit to the TMLR audience in its current state.

**Claims And Evidence:**

The paper considers the failure of Adam and RMSProp when applied to continual learning problems. All three reviewers are critical, as (i) the failure of Adam for continual learning is already known [1]; (ii) the evaluation is limited to simplistic use cases, for which neither of the two was designed; and (iii) the scope if the paper is rather vague, being neither a full review paper nor a deep technical evaluation of a problem, whose relevance is not clearly motivated.





_____
[1] Re-evaluating continual learning scenarios: A categorization and case for strong baselines, Yen-Chang Hsu et al., 2018.

**Resubmission Of Major Revision:**

The authors may consider submitting a major revision at a later time.